# Effects of Testing Temperature on the Serration Behavior of an Al–Zn–Mg–Cu Alloy with Natural and Artificial Aging in Sharp Indentation

**Hiroyuki Yamada [1],*, Tsuyoshi Kami [2] and Nagahisa Ogasawara [1]**

1   Department of Mechanical Engineering, National Defense Academy, 1-10-20 Hashirimizu,
    Yokosuka-shi 239-8686, Kanagawa, Japan; oga@nda.ac.jp
2   Graduate School of Science and Engineering, National Defense Academy, 1-10-20 Hashirimizu,
    Yokosuka-shi 239-8686, Kanagawa, Japan; kamimechnda@gmail.com
*   Correspondence: ymda@nda.ac.jp; Tel.: +81-46-841-3810

**Abstract:** Serration phenomena, in which stress fluctuates in a saw-tooth shape, occur when a uniaxial test is performed on an aluminum alloy containing a solid solution of solute atoms. The appearance of the serrations is affected by the strain rate and temperature. Indentation tests enable the evaluation of a wide range of strain rates in a single test and are a convenient test method for evaluating serration phenomena. Previously, the serrations caused by indentation at room temperature were clarified using strain rate as an index. In this study, we considered ambient temperature as another possible influential factor. We clarify, through experimentation, the effect of temperature on the serration phenomenon caused by indentation. An Al–Zn–Mg–Cu alloy (7075 aluminum alloy) was used as the specimen. The aging phenomenon was controlled by varying the testing temperature of the solution-treated specimen. Furthermore, the material properties obtained by indentation were evaluated. By varying the testing temperature, the presence and amount of precipitation were controlled and the number of solute atoms was varied. Additionally, the diffusion of solute atoms was controlled by maintaining the displacement during indentations, and a favorable environment for the occurrence of serrations was induced. The obtained results reveal that the variations in the serrations formed in the loading curvature obtained via indentation are attributed to the extent of interaction between the solute atoms and the dislocations.

**Keywords:** indentation; serration; temperature; strain rate; dislocation; artificial aging; solid solution; loading curvature; aluminum alloy

## 1. Introduction

A phenomenon called serration—stress fluctuations in a saw-tooth shape—occurs when a uniaxial test (e.g., tensile test) is performed on aluminum alloys containing solute atoms in the solid solution [1–4]. Dynamic strain aging gives rise to the Portevin–Le Chatelier effect [3,5,6]. One of the manifestations of this effect is the serration that occurs when dislocations are pinned or released from the atmosphere of a solute atom. For example, in Al–Mg alloys (5000 series aluminum alloys), significant serrations are generated, as well as a strain pattern similar to a Luder's band on the surface of the alloy, thereby impairing its appearance. Many studies on the serration phenomena have been conducted, primarily involving the 5000 series aluminum alloys. Existing studies identified serration behavior as an interaction between dislocations and the solute Mg [3,5,6]. This phenomenon was reported to be affected by strain rate and temperature because the velocity of dislocation motion and the diffusion rate of Mg are affected by the strain rate and the temperature, respectively [3,5].

Previous studies have demonstrated that the serration behavior varies with change in the strain rate. The classifications of serrations are briefly summarized below [7]. Figure 1 shows an example of variations in the serration behavior with a change in strain rate [8]. It shows an A-type stress fluctuation that repeatedly rises and falls (the wavy stress fluctuation at a relatively high strain rate, $\dot{\varepsilon}_A$) and B-type saw-tooth-like stress fluctuations generated at intermediate strain rate ($\dot{\varepsilon}_B$, $\dot{\varepsilon}_{A+B}$). Serrations sometimes occur as a result of a combination of behaviors. For example, an A + B-type—a combination of A and B-type stress—has been reported. The serrations observed at low strain rates ($\dot{\varepsilon}_C$) are classified as C-type stress (the $C_A$ and $C_B$ types depending on the frequency of fluctuation), in which the fluctuations are irregular. At a strain rate above $\dot{\varepsilon}_A$ or below $\dot{\varepsilon}_C$, serrations do not occur, rather, a smooth stress–strain relationship is obtained.

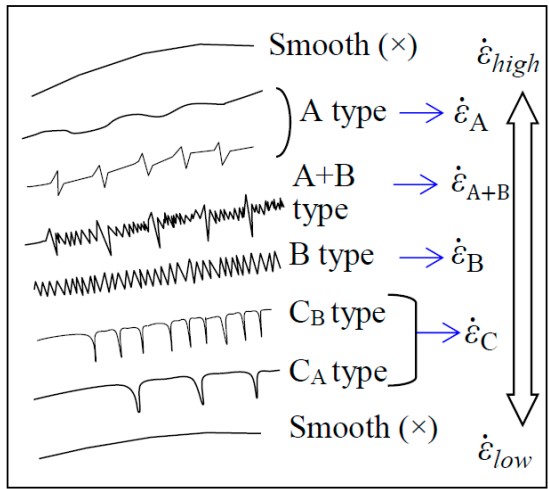

**Figure 1.** Examples of different serration behaviors for different strain rate regimes [8].

To date, serrations have been evaluated primarily by uniaxial assessments such as compression and tensile tests. The strain rate in the uniaxial test is defined by the following equation:

$$\dot{\varepsilon}_u = \frac{d_{\varepsilon u}}{d_t},\tag{1}$$

where $\dot{\varepsilon}_u$ is the strain rate and $\varepsilon_u$ is the strain during the uniaxial test. However, present authors [8] have, through experiment, proved that serration phenomena can be evaluated through indentation tests. Through continuous measurement of the load and the corresponding displacement, while loading and unloading in an indentation test, some mechanical properties that cannot be obtained through hardness tests could possibly be evaluated [9]. Therefore, the indentation test is also a vital non-destructive test for metals. The mechanical properties obtained through indentation test are evaluated using the loading curvature $C$, as shown below:

$$P = Ch^2,\tag{2}$$

where $P$ and $h$ are the load and displacement during indentation, respectively. The loading curvature is given by the following relational expression:

$$C = f\left(E,\ Y,\ n,\ \alpha,\ T,\ \dot{\varepsilon}_i\right),\tag{3}$$

where $E$ is Young's modulus, $Y$ the yield stress, $n$ the work hardening index, $\alpha$ the indenter angle, $T$ the temperature, and $\dot{\varepsilon}_i$ the strain rate of the indentation. Doerner and Nix [10] proposed the following

empirical equation for determining the value of $\dot{\varepsilon}_i$ using a triangular pyramid indenter with the same area to depth ratio as the Vickers pyramid:

$$\dot{\varepsilon}_i = k\left(\frac{\dot{h}}{h}\right), \tag{4}$$

where $k$ is the material constant and $\dot{h}$ is the displacement velocity. This equation does not include the effect of indenter angle. It should be noted here that the concepts of the representative stress ($\sigma_r$) and strain ($\varepsilon_R$) for indentation analysis in order to normalize the load–displacement curves [11–13]. In the indentation test, $\sigma_r$ is the flow stress of the uniaxial test at a particular strain $\varepsilon_r$. The representative strain is given the formula [13]:

$$\varepsilon_R = 0.0638 \; \cot \; \alpha, \tag{5}$$

where $\alpha$ is a half-apex angle. For example, $\varepsilon_R$ is approximated as 0.023 when using a conical indenter with a half-apex angle of 70.3° (described later). Therefore, Equation (4) expresses the strain rate at this representative strain value.

The $\dot{\varepsilon}_i$ has the same dimension as $\dot{\varepsilon}_u$, however, the definition of strain rate for indentation tests differs from that of uniaxial tests. The strain rate of indentation is distributed inside the test materials in a complex manner [14]. Thus, existing studies have proposed the concept of effective strain rate, $\dot{\varepsilon}_e$, to consider the effect of the distribution of the strain rate on the indentation [15–17]. The effective strain rate is given by the formula:

$$\dot{\varepsilon}_e = \beta\left(\frac{\dot{h}}{h}\right), \tag{6}$$

where $\beta$ is a material constant. Equations (4) and (6) have the same form. However, it has been shown that the value of $\beta$ correlates the strain rate in indentation with that in uniaxial tests [15–17].

Previous studies [8,18] indicated that the serration phenomenon in indentation could possibly be evaluated using the concept of effective strain rate. Indentation is performed using a sharp indenter, hence, a complicated deformation field is generated in the test material, whose deformation mechanism has only recently been clarified [19]. Through the indentation tests, it was also discovered that there is a test evaluation limit called critical strain [19] and that serrations could be used as an index to evaluate this effect [18]. However, these previous studies were conducted only in an ambient temperature environment, hence, the effect of temperature on the serration behavior during indentation was not investigated. There are many unknown factors that could affect the behavior of the serrations obtained from indentation.

Until now, the strengths of most metals are evaluated through uniaxial tensile tests. However, next-generation metals are expected to have micro- and nano-scale properties. Therefore, there is a need to adopt such tests as an indentation test that can non-destructively evaluate the strength of a small area with accuracy comparable to that of uniaxial tests. In this study, we extend our previous work [8,18] to clarify the effects of testing temperature on the serration behavior during indentation tests. The microstructural changes in the Al–Zn–Mg–Cu alloy (7075 aluminum alloy) due to natural and artificial aging were employed [20]. In addition, indentation was established as a new method of evaluating material properties through the evaluation of the serration behavior related to the microstructure.

## 2. Materials and Methods

### 2.1. Specimen

A 7075 aluminum alloy (hereafter referred to as the 7075 alloy) specimen was used in this study. Table 1 lists the chemical composition of the 7075 alloy. The dimensions of the cylindrical specimen were 40 mm (diameter) and 40 mm (height), and the end face was finished by lathing. The solution

treatment was conducted at 753 K for 3600 s, followed by water-cooling. Indentation tests were performed on this solution-treated specimen.

**Table 1.** Chemical composition of the investigated 7075 alloy (wt%).

| Alloy | Si | Fe | Cu | Mn | Mg | Cr | Zn | Ti | Al |
|-------|------|------|-----|------|-----|------|-----|------|------|
| 7075 | 0.09 | 0.19 | 1.6 | 0.04 | 2.6 | 0.20 | 5.6 | 0.02 | Bal. |

*2.2. Indentation*

2.2.1. Testing Conditions

A universal testing machine (Instron, series 5982, Norwood, MA, USA) attached with a jig was used for the indentation. Approximately 1 mm was loaded into the lathe-machined surface of the specimen as milli-indentation. A conical indenter (Figure 2a) made of a WC–Co superalloy was used. The radius of the tip (referred to as the roundness) of the indenter was 8.63 μm. The indenter angle was measured to be 141.02° using a laser microscope. This angle is almost equal to that of a conical indenter (140.6°) that has the same indentation projection area at the same indentation depth as those of the Berkovich indenter (a triangular pyramid with a ridge angle of 115°; see Figure 2b). In the existing studies, the effects of the roundness of indenters up to the initial stage of indentation (approximately 1 μm) were reported, where the roundness of the indenter was approximately 10 μm. However, the effect was obtained to be negligible when the indentation was higher than 1 μm [21]. In this study, we assumed an indentation of 1 mm. Therefore, the effect of the roundness of the indenter was minimal.

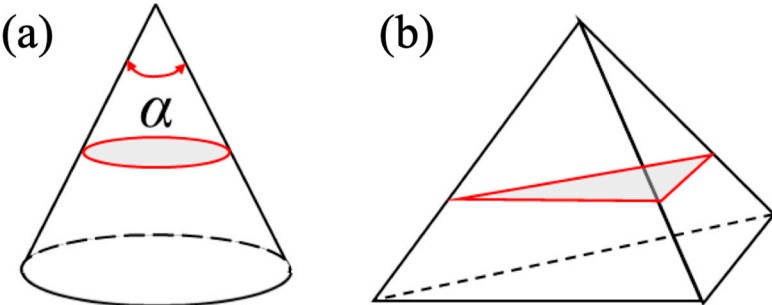

**Figure 2.** Schematic diagram of a cone-type indenter: (**a**) conical type and (**b**) Berkovich (triangular pyramid) type. Here $\alpha$ is the indenter angle and the shaded section is the projection area.

In this study, the temperature at which the Guinier–Preston (GP) zones and the $\eta'$ phase precipitate in 7075 alloys was adopted as the testing temperature (described later). First, a temperature of 343 K at which GP zones have been confirmed to precipitate (or nucleate) [20,22] was chosen. The $\eta'$ phase has often precipitated under the condition of aging at 393 K for 24 h (T6 temper). In this study, however, a temperature of 443 K at which the $\eta'$ phase has been reported to precipitate within a short time (600 to 3600 s) [20] was adopted. Figure 3 shows a schematic diagram of the indentation test at high temperatures. A home-built electric furnace was used. A thermocouple was attached to the specimen surface and was controlled to a predetermined temperature using a temperature controller (CHINO, SY2111, Tokyo, Japan). The arrival times at 343 and 443 K were approximately 1800 and 3600 s, respectively. The specimen was held for 900 s at each of the temperatures, after which indentation was performed. Furthermore, the specimen was held at 77 K in a liquid nitrogen where the low-temperature indentation was performed. At this temperature, aging takes place at a very low rate. Unlike the high-temperature test, the low-temperature indentation test was performed in a container using waterproof paper. The specimen was also indented at room temperature (293 K), thus, the test was performed in four different temperature environments. For each of the temperature conditions,

the test preparation was initiated within 300 s of the solution treatment. Therefore, the effect of natural aging was small, except at room temperature.

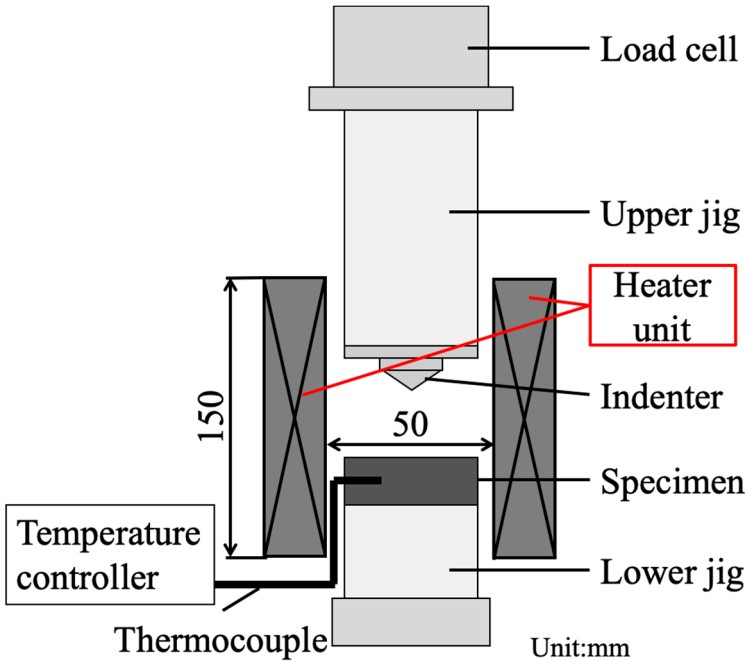

**Figure 3.** A schematic diagram of milli-indentation at high temperatures (343 and 443 K).

### 2.2.2. Indenter Control

The control methods for indenters can be classified into two: loading rate and displacement rate controls. In this study, a commercial universal testing machine was used, hence, displacement rate control was employed. During indentation, the strain rate was varied by varying the displacement rate (see Figure 4). Previous studies have shown that serrations are being affected by the diffusion of Mg in the solid solution [5]. The time required for Mg atoms in the solid solution to sufficiently pin stagnant dislocations is given by the following equation [23,24]:

$$t_a = \left(\frac{C_1}{3C_0}\right)^{\frac{3}{2}} \frac{kTb^2}{3DU_m},\tag{7}$$

where $C_1$ is the concentration of solid solution atoms required for serrations to occur, $C_0$ the concentration of solid solution atoms in the material, $k$ the Boltzmann's constant, $T$ temperature, $D$ the diffusion rate, and $U_m$ the binding energy between the solid solution atoms and dislocations. When clusters and GP zones are formed by aging, the amount of Mg in the solid solution decreases, and accordingly, $C_0$ decreases. $D$ also increases as temperature increases [25,26]. It is difficult to measure the diffusion rate of solute atoms at low temperatures. Therefore, we predicted using the following equation:

$$D = D_0 \exp\left(-\frac{Q}{RT}\right),\tag{8}$$

where $D_0$ is the frequency factor, $Q$ is the activation energy and $R$ is the gas constant. Table 2 shows the prediction of the diffusion rate of solute atoms in aluminum at the testing temperature [27–29]. Hence, to promote the diffusion of solute atoms, a constant displacement is maintained. As shown in Figure 4, the displacement was held constant at two different values, for 20 s each, during the indentation test.

**Table 2.** Prediction of the diffusion rate of solute atoms in aluminum at the testing temperature using Equation (8).

| Solute Atom | $D_0$ (m$^2$/s) | $Q$ (kcal/mol) | Testing Temperature (K) | $D$ (m$^2$/s) | Reference |
|---|---|---|---|---|---|
| Zn | $1.77 \times 10^{-5}$ | 28.0 | 77 | $7.74 \times 10^{-85}$ | [27] |
| | | | 293 | $2.47 \times 10^{-26}$ | |
| | | | 343 | $2.71 \times 10^{-23}$ | |
| | | | 443 | $2.85 \times 10^{-19}$ | |
| Mg | $6.23 \times 10^{-6}$ | 27.5 | 77 | $7.11 \times 10^{-84}$ | [28] |
| | | | 293 | $2.05 \times 10^{-26}$ | |
| | | | 343 | $1.98 \times 10^{-23}$ | |
| | | | 443 | $1.77 \times 10^{-19}$ | |
| Cu | $1.5 \times 10^{-5}$ | 30.2 | 77 | $3.81 \times 10^{-91}$ | [29] |
| | | | 293 | $4.81 \times 10^{-28}$ | |
| | | | 343 | $9.14 \times 10^{-25}$ | |
| | | | 443 | $1.99 \times 10^{-20}$ | |

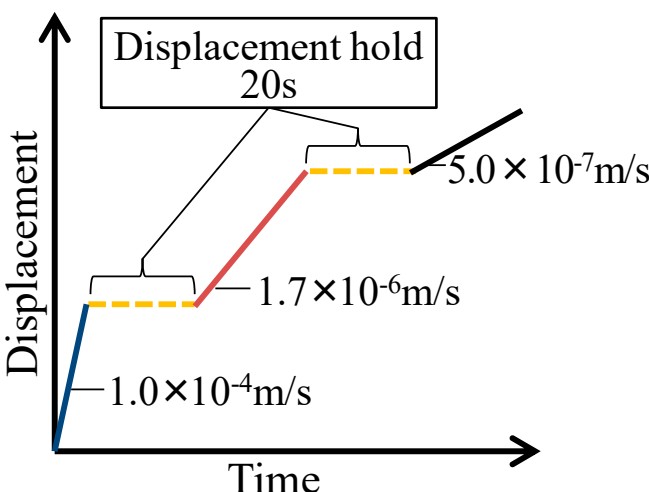

**Figure 4.** A schematic diagram of the time history of displacement.

## 3. Results

Figure 5 shows the load–displacement relationships for all the tests at different temperatures. For the tests at temperatures above room temperature, the load increased with an increase in temperature regardless of the change in the displacement rate. This increase in the load is attributed to the precipitates formed by artificial aging. At 77 K, a decrease in the displacement rate lowered the increase rate of the load as the displacement increased as compared to the other temperatures.

The effective strain rates for the indentations were calculated using Equation (6). Herein, $\beta = 0.1$ was used, based on previous studies [8,18]; the effective strain rate–displacement relationship is shown in Figure 6. The effective strain rate under displacement rate control, given by Equation (6), decreased as the displacement increased. The indentations were performed at three different rates by varying the indenter speed (see Figure 4). A wide range of effective strain rates (from $10^{-4}$ to $10^0$ s$^{-1}$) was obtained during the indentations. There was no significant difference in the effective strain rate, even when the testing temperature changed. Aluminum alloys are known to have high strength–strain rate sensitivity at lower temperatures [30,31]. This implies that at extremely low temperatures, the strength of aluminum alloys decreases as the strain rate decreases. Therefore, during the cryogenic indentation, the decrease in the increase rate of the load was as a result of the decrease in the effective strain rate with increasing displacement.

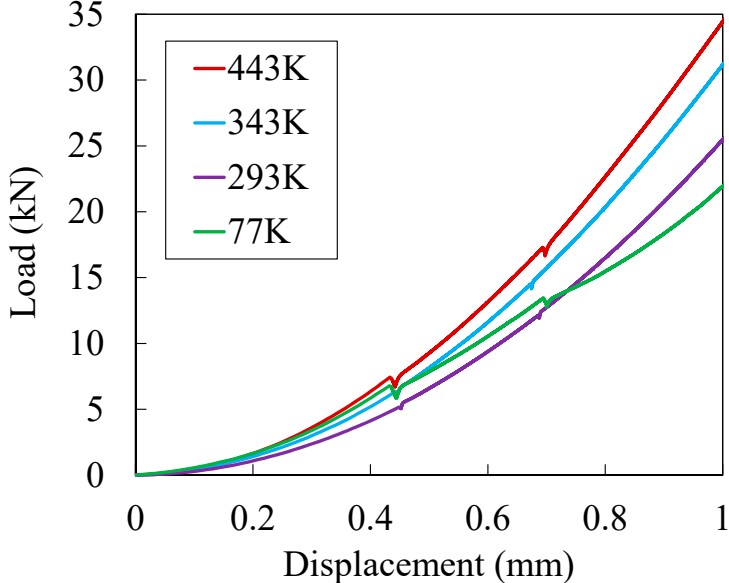

**Figure 5.** Load–displacement curves at 77, 293, 343, and 443 K. The change in displacement rate during indentation is shown in Figure 4.

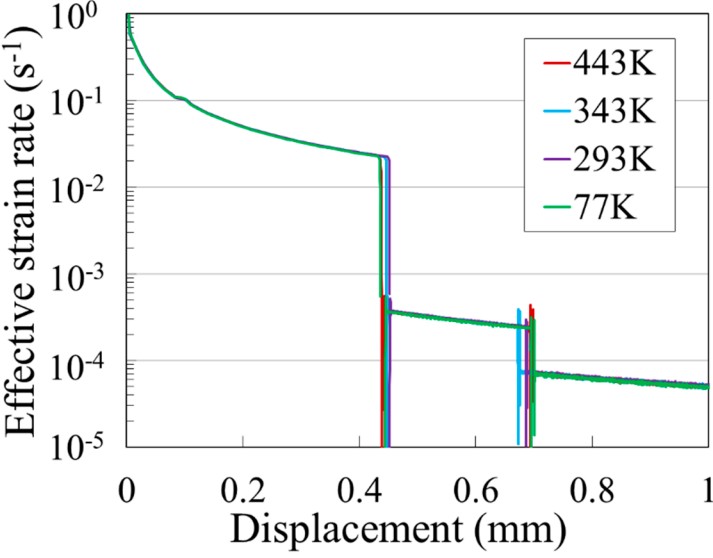

**Figure 6.** Effective strain rate–displacement relationship at each testing temperature.

We calculated the loading curvature-displacement relationship from the load–displacement relationship using Equation (2) (see Figure 7). The effect of temperature and strain rate on the indentation was confirmed by the change in the loading curvature. When testing temperatures greater than the room temperature, the loading curvature was observed to increase as the temperature increased regardless of the change in the displacement rate. At 293 and 343 K, an increase in the loading curvature was observed after holding as compared with that before holding. By contrast, there was a decrease in the loading curvature after holding for the test conducted at 77 and 443 K. At 77 K, not only the time in which the displacement rate increases but also the loading curvature decreases with increasing displacement.

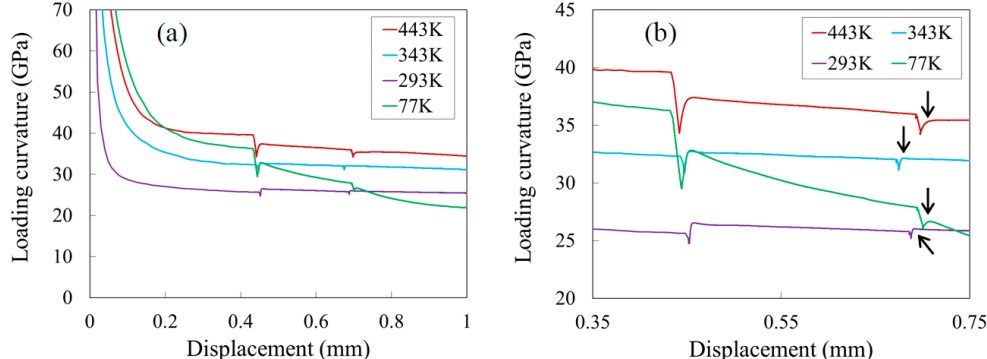

**Figure 7.** (**a**) Loading curvature-displacement curves at 77, 293, 343, and 443 K; (**b**) enlarged view. The change in displacement rate during indentation is as shown in Figure 4.

## 4. Discussion

### 4.1. Effect of Aging on the Material Strength

The precipitation process of 7075 alloys established in previous research is as follows: [32–34].

$$\text{Supersaturated solid solution (ssss)} \rightarrow \text{vacancy-rich clusters (VRC)} \rightarrow \text{GP zone} \rightarrow \eta' \rightarrow \eta. \quad (9)$$

A cluster or GP zone is an aggregate of atoms with a diameter of the order of nanometres. Herein, $\eta'$ denotes the metastable phase, whereas $\eta$ denotes the stable phase. Clusters and GP zones are formed during the natural and artificial aging, indicating an increase in material strength. After additional aging, $\eta'$ precipitates and the strength of the material reaches its climax (peak aged). However, as $\eta'$ continues to grow and $\eta$ starts to precipitate, there is a decrease in the strength of the sample (over-aged). Therefore, increasing the testing temperature increases the strength as a result of the formation of precipitates until the peak-age condition is reached. The amount of solute atoms is also decreased. The succeeding sections discuss each testing temperature based on the above findings.

#### 4.1.1. Temperature of 77 K

In [15], when the load was held during indentation at room temperature, the loading curvature after holding was obtained to be higher than that before the holding. This is attributed to the fact that an amount of solute atoms in solid solution diffuses into the dislocations during the holding period, thereby, causing the pinning of dislocations. At 77 K, which is a very low-temperature environment, the above-mentioned precipitation process was not observed and the sample probably remained in a solid solution. Therefore, solute atoms were in the solid solution during the indentation. However, there was no increase in the loading curvature after holding. This indicates that solute atoms have not segregated (or diffused) to the dislocations and pinned them in this low-temperature environment because solute atom diffusion is very slow as shown in Table 2, i.e., no time for diffusion.

#### 4.1.2. Temperature of 293 and 343 K

In agreement with the results obtained in a previous study [18], an increase in the value of the loading curvature after holding was observed at 293 and 343 K as a result of dislocation pinning caused by the diffusion of solute atoms into the dislocations. At 343 K, the GP zone formed inside the material as a result of aging. However, a sufficient number of solute atoms to cause dislocation pinning was expected to be retained in the solution.

#### 4.1.3. Temperature of 443 K

In contrast to the results obtained in the tests conducted at 293 and 343 K, at 443 K, the value of the loading curvature after holding was smaller than that before holding. This could be attributed to

the fact that the amount of solute atoms must have been significantly reduced by aging, hence, the dislocation pinning effect was less likely to occur.

### 4.2. Effect of Testing Temperature on the Serration Behavior

Figure 8 shows an enlarged view of the region of the curve that is indicated by the arrow in Figure 7b as a means to investigate the details of the loading curvature-displacement relationship. Serrations were observed at 293 and 343 K but not at 77 and 443 K. The effective strain rate was approximately $7 \times 10^{-4}$ s$^{-1}$ at all testing temperatures (see Figure 6). It has been stated that the serrations observed in uniaxial tests were affected by the strain rate and the testing temperature. To discuss the effect of testing temperature on the serration behavior observed in this study, the testing temperatures related to indentation were investigated.

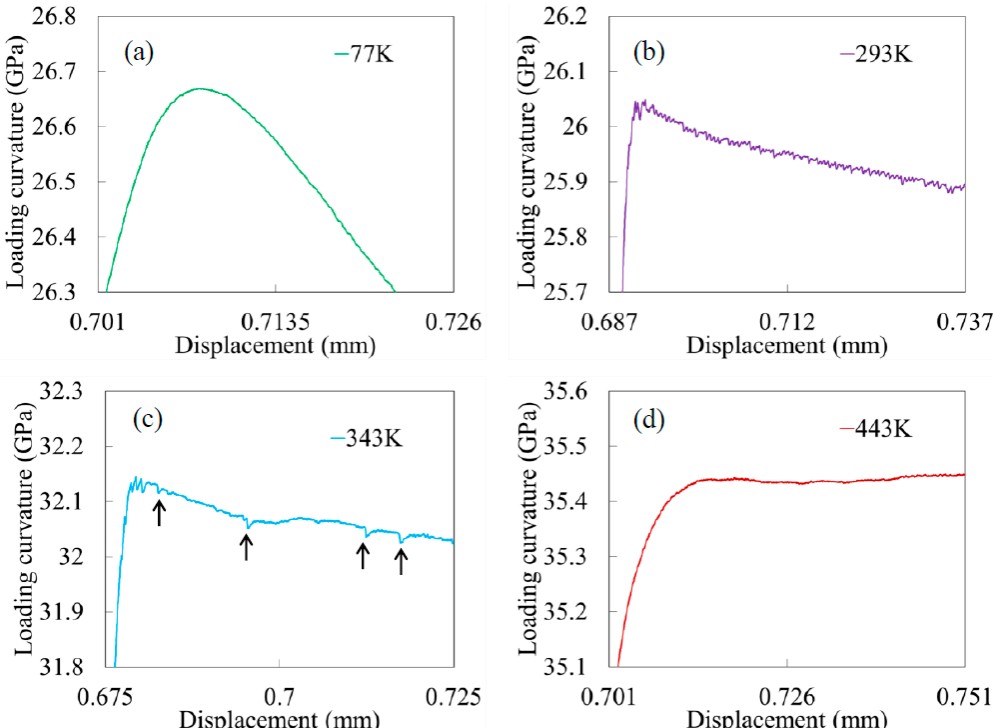

**Figure 8.** Enlarged view of Figure 7b: (**a**) 77 K, (**b**) 239 K, (**c**) 343 K, and (**d**) 443 K.

### 4.2.1. Temperature of 77 K

At 77 K, no fluctuation was observed in the loading curvature, as shown in Figure 8a, hence, serrations did not occur at this temperature. Because serrations occur at room temperature, it is assumed that the effect was not due to strain rate. This indicates that aging does not occur at very low temperatures. It also shows that there was barely any interaction between the dislocations and solute atoms.

### 4.2.2. Temperature of 293 K

As shown in Figure 8b, there was a significant fluctuation in the loading curvature. The occurrence of B-type serrations was confirmed (see Figure 1). There was a large number of solid solution atoms in the sample as 293 K corresponds to the early stage of aging. Consequently, there was a tendency for interaction between dislocations and solute atoms in solid solution to occur. Thus, the occurrence of a significant number of serrations was confirmed.

### 4.2.3. Temperature of 343 K

At 343 K, as shown by the arrow in Figure 8c, the interval at which the loading curvature dropped is greater than that at 293 K (see Figure 8b). Therefore, A- and B-type serrations (see Figure 1) were confirmed to have occurred. As the strain rate was constant, other causes of change in the serration phenomenon were observed. These are inferred to be the formation of GP zones, thus, a decrease in the number of solute atoms, and the increase in testing temperature. When the amount of solute atoms decreases, the chance of interaction with dislocations decreases, hence, serrations barely occur.

### 4.2.4. Temperature of 443 K

No serration was observed at 443 K because of the formation of $\eta'$ and the increase in the strength of the material. The amount of solute atoms was, therefore, greatly reduced compared with other testing temperatures. Thus, the interaction between dislocations and solid solution atoms was less likely to occur. In addition, the diffusion rate of solute atoms at 443 K is higher than that at 343 K, hence, it is inferred that the remaining solid solution solute atoms were pinned to dislocations and deviates from the conditions at which serrations could occur. Therefore, it is believed that the interaction between dislocations and solid solution atoms does not appear in the loading curvature.

## 5. Conclusions

In this study, to clarify the effect of temperature change on the resulting serrations during indentation tests, we performed milli-indentations on an aluminum alloy (7075 alloy) at various temperatures. The serration phenomenon during indentation was varied by controlling the number of precipitated phases based on the effect of natural and artificial aging. This variation was as a result of the interaction between dislocations and the solid solution atoms observed under the different testing temperatures and strain rate on indentation, similar to that observed in previously reported uniaxial test results. Therefore, the serration phenomenon can be investigated via sharp indentation tests, which is considered valuable as a non-destructive testing technique for evaluating the dynamic strain aging of next-generation metals.

This study focused on the interaction between dislocations and the number of solute atoms based on the varying temperature and strain rate in indentation tests. Thus, the effect of the number of solute atoms and the testing temperature cannot be separated. Therefore, qualitative evaluations (e.g., microstructure evaluation by transmission electron microscopy or X-ray diffraction) to study the conditions that may separate these effects is recommended for future studies.

**Author Contributions:** H.Y. and T.K. conceived, designed, and performed the experiments; N.O. considered from research results; H.Y. and T.K. drafted this paper. All authors have read and agreed to the published version of the manuscript.

**Funding:** This research was funded by The Light Metal Educational Foundation, Inc., Japan.

**Acknowledgments:** The authors would like to thank Hikaru Ootani for assisting with the conduction of the indentation test.

**Conflicts of Interest:** The authors declare no conflict of interest.

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
