# Peer review of "Effects of Testing Temperature on the Serration Behavior of an Al–Zn–Mg–Cu Alloy with Natural and Artificial Aging in Sharp Indentation"

_metals, doi:10.3390/met10050597_

Round 1
Reviewer 1 Report
A help of native English is highly recommended in order to correct the grammar mistatakes
14: Does "Indentation" mean a hardness test? If so, it will be more appropriate to use the "Indentation test"
17: The correct statement will be the „ambient temperature” instead of the „room temperature”
43: Each one of the cited sources should be discussed individually and explicitly to demonstrate their significance to your study. It would better to refer to the authors' surnames, and then state in one or two sentences what they claim, what evidence they provide to support their claims, and how you evaluate their work.
54: In the description of figure 1 there is temperature mentioned. However, the authors do not provide temperature.
60: From the description it is not clear what method the authors use in the research. Is it hardness test or compression test done by means of indenter?
61: Please explain, what is the C curve
79: Can you provide the literature concerning "the deformation mechanism of which has only recently been clarified". Perhaps it is literature [13]?
87: Please check whether the term artificial aging is correct in this case. Temperature studies indicate that the authors used both artificial and natural aging.
95: Although ks is an SI unit, I suggest changing to 1 h, or 3600 s
122: How was the thermocouple attached to the sample? Please present the appropriate drawing.
123: Please explain how much time passed between reaching the desired temperature of the beginning of the test. Please present temperature-time curves from the beginning till the end of the test.
125: Please explain whether the same test stand was used for the 77 K sample test. If not, please describe the test stand.
127: Ambient temperature and 293 K seems to be the same temperature why they were mentioned separately?
130: It is good practice to show the sample after the test.
169: Please describe how much time passed after solutioning. Has natural aging occurred in the sample?
192: Please check whether the GP zones are formed during artificial ageing. According to my knowledge they are formed during natural aging process.
228: Please explain why there are such high values presented in the vertical axis of most of the graphs?
Reviewer 2 Report
See attached file

Reviewer 3 Report
Dear Authors
1) In my opinion the conclusion provides mostly descriptive information. Discussion of process kinetics in comparison with previous studies (with relevant references) would be useful.
2) The role of diffusion is not absolutely clear and an additional discussion neads to comfirm this point. The XRD profile of the Mg befor and after the heat treatment would be a good test showing changes in Mg diffusion.
3) Please read about the other condition of aging 7075 alloy:
Kaczmarek Ł., Stegliński M., Sawicki J., Świniarski J., Batory D., Kyzioł K., Kołodziejczyk Ł., Szymański W., Zawadzki P., Kottfer D.: Optimization of the heat treatment and tribological properties of 2024 and 7075 aluminium alloys, Archives of Metallurgy and Materials, Vol.58, issue 2, 2013, 535-540
4) Some formal mistake
Line 85
“In this study, we extend our previous work [……] to clarify the effects of testing temperature on….” - insert reference [.....]
Line 127
“….including 293 K and room temperature.” – 273 is a room temperature – so put in bracket (room temperature)
Round 2
Reviewer 2 Report
The revised version of the manuscript has improved as compared to the first version of the paper. However, it is the present reviewer’s clear opinion, that it is still not satisfactory in several respects, and needs further extensive revisions before possibly becoming acceptable for publication. This applies both to the language as well as elements of the content.
A marked copy of the manuscript, which exemplifies language issues as well as indicating strange and unclear formulations and with some suggestions for changes is attached. NOT EXHAUSTIVE, the authors need to care for yet another careful revision of the whole manuscript, mainly in terms of the language (grammar), but also to some elements of the content.

Round 3
Reviewer 2 Report
Cf. attached file.
